



# Gas exchange velocities $(k_{600})$, gas exchange rates $(K_{600})$, and hydraulic geometries for streams and rivers derived from the *NEON Reaeration field and lab collection data product* (DP1.20190.001)

Kelly S. Aho[1], Kaelin Cawley[2], Robert Hensley[2], Robert O. Hall, Jr.[3], Walter Dodds[4], Keli Goodman[2]

[1]Department of Earth and Environmental Science, Department of Integrative Biology, Michigan State University, East Lansing, MI 48824, USA
[2]National Ecological Observatory Network, Battelle, 1685 38th St. #100, Boulder, CO 80301, USA
[3]Flathead Lake Biological Station, University of Montana, Polson, MT 599111, USA
[4]Division of Biology, Kansas State University, Manhattan, KS, 66506, USA

*Correspondence to*: Kelly S. Aho (kellyaho@msu.edu)

**Abstract.** Air-water gas exchange is essential to understanding and quantifying many biogeochemical processes in streams and rivers, including greenhouse gas emissions and metabolism. Gas exchange depends on two factors, which are often quantified separately: 1) the air-water concentration gradient of the gas and 2) the gas exchange velocity. There are fewer measurements of gas exchange velocity compared to concentrations in streams and rivers, which limits accurate

characterization of air-water gas exchange (i.e., flux rates). The National Ecological Observatory Network (NEON) conducts SF₆ gas-loss experiments in 22 of their 24 wadeable streams using standardized methods across all experiments and sites, and publishes raw concentration data from these experiments on the NEON data portal. NEON also conducts NaCl injections that can be used to characterize hydraulic geometry at all 24 wadeable streams. These NaCl injections are conducted both as part of the gas-loss experiments and separately. Here, we use these data to estimate gas exchange and

water velocity using the reaRates R package. The dataset presented includes estimates of hydraulic parameters, cleaned raw concentration SF₆ tracer-gas data (including removing outliers and failed experiments), estimated SF₆ gas loss rates, normalized gas exchange velocities ($k_{600}$, $m\ d^{-1}$), and normalized depth-dependent gas exchange rates ($K_{600}$, $d^{-1}$). This dataset provides one of the largest compilations of gas loss experiments (n = 339) in streams to date. This dataset is unique in that it contains gas exchange estimates from repeated experiments in geographically diverse streams across a range of

discharges. In addition, this dataset contains information on the hydraulic geometry of all 24 NEON wadeable streams, which will support future research using NEON aquatic data. This dataset is a valuable resource that can be used to explore both within- and across-reach variability in the hydraulic geometry and gas exchange velocity in streams.

## 1 Introduction

Air-water gas exchange contributes to many aquatic processes in streams and rivers, including greenhouse gas emissions (Liu et al., 2022; Rocher-Ros et al., 2023), aquatic metabolism (Aristegi et al., 2009; Hall et al., 2016; Hall and Hotchkiss, 2017), and reoxygenation rates after wastewater discharge (O'Connor and Dobbins, 1958). Despite this importance, gas exchange can be difficult to measure and model (Churchill et al., 1964; Hornberger and Kelly, 1975; Rathbun, 1977; Ulseth et al., 2019). According to Fick's Law of Diffusion, gas flux across the air-water boundary depends

on the concentration gradient of the gas and the gas exchange velocity ($k$, $m\ d^{-1}$):

$$Flux = k\big([gas]_{dissolved} - [gas]_{equilibrium}\big), \qquad (1)$$



Where $[gas]_{dissolved}$ is the concentration of the gas of interest and $[gas]_{equilibrium}$ is the concentration of the gas
at equilibrium with the atmosphere.

In streams and rivers, measurements of gas concentrations are more readily available than estimates of $k$; resultantly, estimates of $k$ are often extrapolated from a few measurements. Several methods exist for assessing $k$, including predictive models (Raymond et al., 2012), models of gas dynamics through time and space in rivers (Appling et al., 2018), and direct measurements with tracers (Hall and Hotchkiss, 2017). Direct measurement of tracer-gas exchange velocities and
modeling based on observed diurnal gas dynamics are likely more accurate for any particular stream or river than more general predictive models (Appling et al., 2018; Hall and Ulseth, 2020; Riley and Dodds, 2013).

Gas exchange velocity is spatiotemporally variable; it is controlled by energy dissipation rate and, therefore, turbulence at the air-water boundary (Zappa et al., 2007). Models that estimate $k$ at broad spatial scales and in low-versus-high gradient streams have found that hydraulic variables (e.g., streambed slope [$S, unitless$], water velocity [$v, m\ s^{-1}$],
mean water depth [$\bar{z}, m$], discharge [$Q, L\ s^{-1}$]) are the dominant controls on variation in $k$ (Churchill et al., 1964; O'Connor and Dobbins, 1958; Rathbun, 1977; Raymond et al., 2012). Although similar models for within-reach temporal variability are not widely available, hydrology is expected to control $k$ locally. Existing reach-scale studies have reported different $k$ responses to $Q$ (Genzoli and Hall, 2016; Maurice et al., 2017; McDowell and Johnson, 2018) and point to the importance of quantifying the variable relationships between $k$ and $Q$ on a site-by-site basis. The dataset presented here leverages a unique
set of tracer-gas experiments conducted by the National Ecological Observatory Network (NEON) that will allow for assessment of within- and across-reach variability in lotic gas exchange in a wide variety of streams.

Tracer-gas experiments are an approach to estimating $k$ in streams and rivers and involve diffusing an inert tracer gas (e.g., sulfur hexafluoride [$SF_6$]) at a constant rate at the upstream end of a stream reach and measuring how concentrations decline downstream of the injection point. Often a conservative salt (e.g., sodium chloride [NaCl] or sodium
bromide [NaBr]) is also injected with the tracer gas to allow for correction of dilution from hydrologic inflows (referred to as "salt correction" hereafter).

Here, we present a continental-scale dataset of gas exchange rates for wadeable streams derived from NEON data. The substantial processing that was required to estimate gas exchange is described in detail below and archived alongside the dataset. The dataset presented here contains estimates of $k_{600}$ and $K_{600}$ for 22 streams and $v$, $\bar{z}$, and at-a-station hydraulic
geometry for 24 streams (Figure 1).

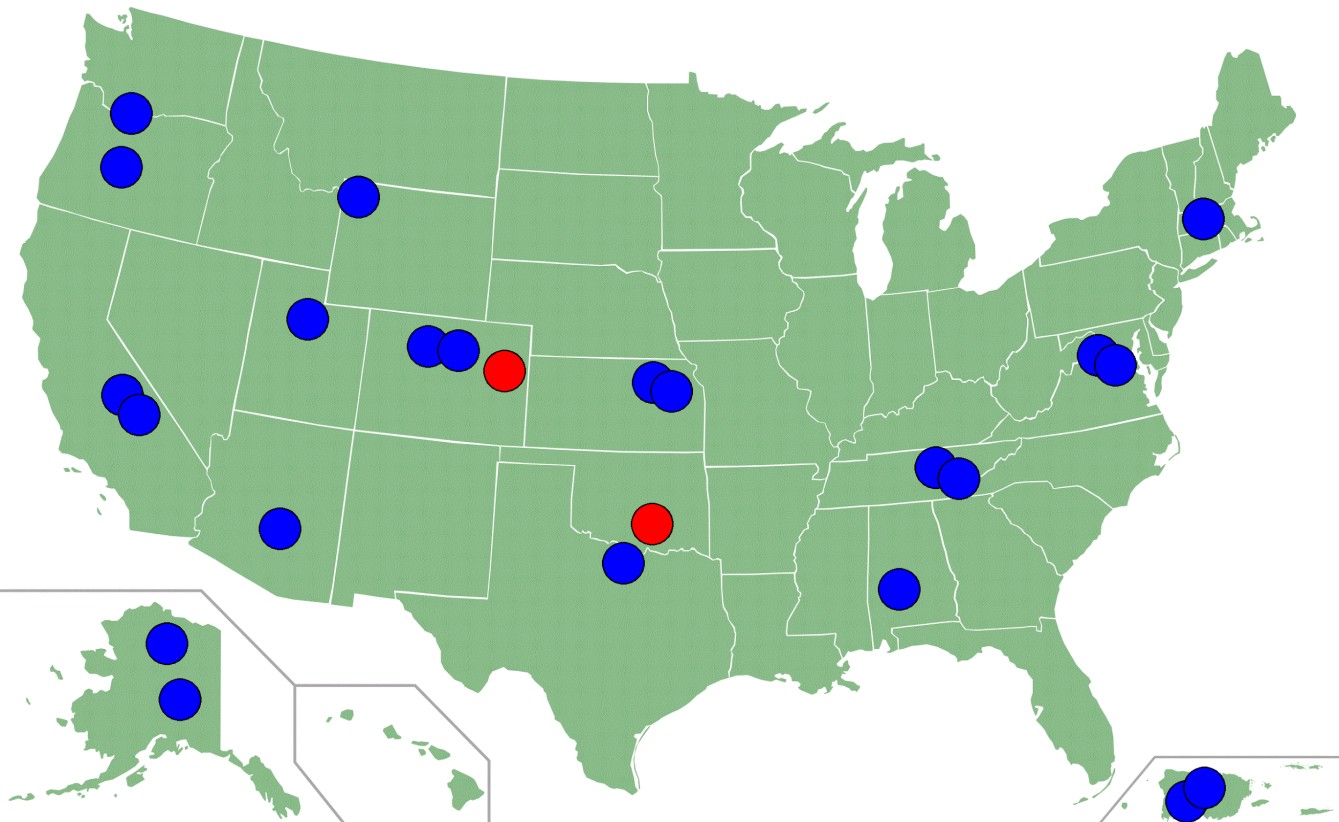

**Figure 1. Map of NEON stream sites. Blue symbols indicate sites where NEON conducts full tracer-gas experiments and, thus, where we were able to estimate $k_{600}$, $K_{600}$, $v$, $\bar{z}$, and at-a-station hydraulic geometry. Red symbols indicate sites where NEON only conducts NaCl slug injections and, thus, where we were able to calculate $v$, $\bar{z}$, and at-a-station hydraulic geometry.**

## 2 Methods

### 2.1 NEON tracer-gas experiments

NEON conducts tracer-gas experiments at 22 stream sites, which are distributed across the United States, from Puerto Rico to Alaska (Figure 1). For information about specific site characteristics, see the NEON website: https://www.neonscience.org/field-sites/explore-field-sites. These experiments entail continuous injections of $SF_6$ and a conservative salt tracer (either NaCl or NaBr) at the upstream end of a $\leq 500$-m stream reach (Figure 2). When NaBr is used as the salt tracer, an additional NaCl "slug" injection is performed, which allows for the subsequent calculation of $v$ and the derivation of $\bar{z}$ from paired $Q$ measurements (via flowmeter or ADCP) and wetted width measurements taken at 30 points along the study reach. Before the injection, NaCl or NaBr are collected at each of the four stations along the study reach; these data can later be used to correct NaCl or NaBr concentrations during the injection for background conditions. Once conductivity during the injection either reaches a plateau (for constant-rate NaCl injections) or returns to background levels (for NaCl slug injections) at the most downstream station, five replicate $SF_6$ and NaCl or NaBr samples are collected at each of the four stations located along the study reach. In addition, high-frequency (0.1-Hz) sensors are deployed to monitor NaCl conductivity at the upstream and downstream end of the study reach (Figure 2). NEON publishes $SF_6$ mixing ratios, NaCl and NaBr concentrations, wetted width data, and conductivity timeseries from these experiments as the Reaeration

field and lab collection data product DP1.20190.001 (NEON, 2024b) and measurements of $Q$ in the Discharge field collection data product DP1.20048.001 (NEON, 2024a). More detailed information on NEON's data collection procedures can be found on their website, http://www.neonscience.org.

90        NEON has conducted tracer-gas experiments 6 - 10 times per year for 6 - 8 years at all 22 sites to capture a range of discharge conditions. Presently, tracer-gas experiments are ceasing at sites with sufficient hydrograph coverage (https://www.neonscience.org/impact/observatory-blog/protocol-change-reaeration-field-and-lab-collection-dp120190001). However, NaCl slug injections will continue to be performed quarterly to collect high-frequency conductivity time-series data that allow for the calculation of $v$ and the derivation of $\bar{z}$ from paired $Q$ and wetted-width measurements. Similarly,
NaCl slug injections are and will continue to be conducted for the two sites where tracer-gas experiments are not collected (Blue River [BLUE], where wide channel widths and high discharges make tracer -gas studies challenging; Arikaree River [ARIK], where long travel times make tracer-gas studies infeasible).

       The dataset presented here represents substantial processing of these published data (i.e., $SF_6$ mixing ratios; NaCl and NaBr concentrations; conductivity timeseries; wetted widths; and measurements of Q) to estimate $k_{600}$ or $K_{600}$ values
using the reaRate R package (Cawley et al., 2024). In addition, this dataset contains estimates of $v$ from NaCl injections, which, as mentioned above, are performed both during tracer-gas experiments and at quarterly intervals at sites where tracer-gas experiments are not conducted or have ceased. Along with the paired $Q$ measurement and the average wetted width for the study reach ($\bar{w}, units: m$), the estimate of $v$ was used to derive $\bar{z}$. The dataset presented here contains estimates of $v$ and $\bar{z}$ and at-a-station hydraulic geometry for all 24 NEON wadeable streams. This dataset provides a large compilation of direct
measurements of tracer-gas experiments and at-a-station hydraulic geometry in small streams across the United States.

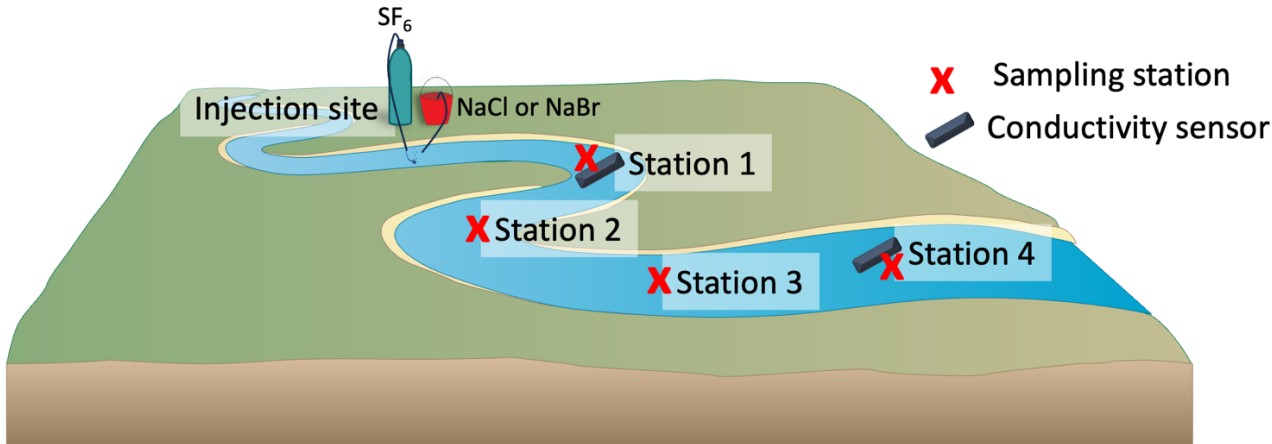

**Figure 2. Diagram of a model study reach for NEON tracer-gas experiments. Each ~500-m study reach comprises an injection site and four downstream sampling stations (Stations 1 – 4). At the upstream injection site, $SF_6$ is diffused into the water column with an air stone and NaCl or NaBr is dripped into the stream. After plateau concentrations are reached at the downstream end of the**
**study reach, injection rates are maintained and field quintuplicate samples for $SF_6$ and NaCl or NaBr concentrations are collected at four downstream stations spaced along the study reach. At the upper and lower stations, conductivity sensors are deployed and used to monitor either 1) when NaCl plateau concentrations are reached (for NaCl continuous injections) or 2) when a NaCl "slug" peaks at each station (for NaBr continuous injections). Before each experiment, stream discharge is measured with a flow meter or ADCP and 30 wetted widths are collected across the study reach. Also, before each injection, background NaCl or NaBr**
**concentrations are collected at all four sampling stations. This diagram uses modified imagery from University of Maryland Center for Environmental Science Integration and Application Network.**

### 2.2 reaRates R package

       The data processing pipeline described below uses the reaRate R package (Cawley et al., 2024). The package estimates $k_{600}$ and $K_{600}$ from data available on the NEON data portal. The package works by fitting an exponential, first-



order decay function to the observed longitudinal decline in published SF₆ and solving for the longitudinal tracer-gas loss rate ($K_d, m^{-1}$):

$$C_x = C_0 e^{-K_d x}, \tag{1}$$

where $C_0$ and $C_x$ are tracer-gas concentrations at the top of the study reach and at a downstream point $x$, respectively, and $K_d$ is the average distance traveled by an SF₆ molecule before it is emitted to the atmosphere. For sites where lateral inflows (e.g., groundwater inputs, overland flow, tributaries) appear to dilute SF₆ concentrations, the ratio of SF₆ to NaCl or NaBr is used to calculate a salt-corrected $K_d$ value. The $K_d$ values can then be converted to the gas exchange velocity for the tracer gas (e.g., $k_{SF_6}, m\ d^{-1}$):

$$k_{SF_6} = \overline{z}\ v\ K_d . \tag{2}$$

This gas-specific $k$ can be normalized to $k_{600}$ using a Schmidt number of 600:

$$k_{600} = k_{SF_6} \left(\frac{600}{Sc_{SF_6}}\right)^{-n}, \tag{3}$$

Where $n$ is the Schmidt number exponent (0.5 for flowing waters) and $S_{c,SF_6}$ is the temperature-dependent Schmidt number for SF₆ at water temperature $T$ in degrees Celsius (Jähne et al., 1987; Raymond et al., 2012; Wanninkhof, 1992):

$$S_{c,SF_6} = 3255.3 - 217.13\ \text{T} + 6.8370\ T^2 - 0.086070\ T^3. \tag{4}$$

Reporting estimates of $k_{600}$ is common; it allows for comparisons with existing work and can be scaled to other gases, including $CO_2$ and $O_2$, using the same approach as in equation 3 with gas-specific, temperature-dependent Schmidt numbers:

$$k_{gas} = k_{600} \left(\frac{Sc_{gas}}{600}\right)^{-\frac{1}{2}}. \tag{5}$$

Some applications (e.g., metabolism, reoxygenation rates) explicitly include water depth in their modeling frameworks, and thus require a depth-dependent gas exchange rate ($K, d^{-1}$). In these cases, a normalized gas exchange rate ($K_{600}$) can be used and is related to $k_{600}$ by dividing by $\overline{z}$ for the upstream reach corresponding to a length of at least $\frac{1}{K_d}$. Using the same scaling relationships shown in equations 3 and 5, $K_{600}$ can be converted to gas-specific $K_{gas}$ estimates.

The reaRate package uses NaCl breakthrough curves (i.e., either from continuous injection or slug injections) to estimate the travel time between the upstream and downstream stations and then calculates $v$ as the distance between stations divided by the travel time. Using the continuity equation, $\overline{z}$ is calculated by dividing $Q$ by $v$ and $\overline{w}$. Finally, $k_{SF_6}$ and $K_{SF_6}$ are calculated from $K_d$, $v$, and $\overline{z}$ (equation 2) and then normalized to $k_{600}$ and $K_{600}$ (equation 3). The reaRates package includes two approaches to estimate $k_{600}$ and $K_{600}$: an un-pooled frequentist approach and a partially pooled Bayesian approach, both of which are described in more detail below. Implementation of the package and all processing described below were conducted in R 4.2.3 (R Core Team, 2023). More information about the package, including details about the individual functions and a processing pipeline, is provided in section 2.3.

**2.3 Data processing**



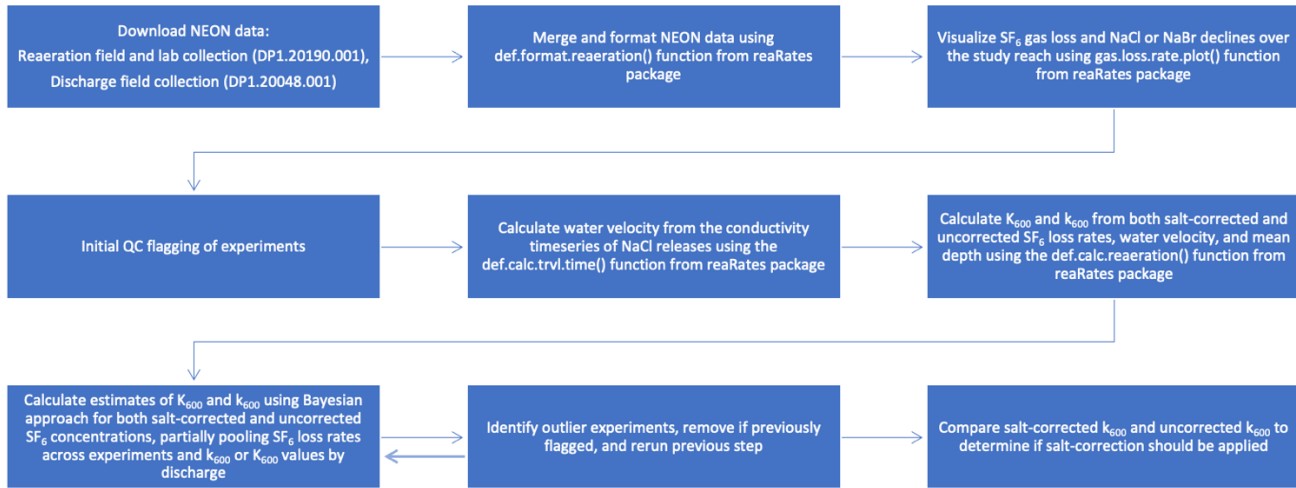

**Figure 3. Overview of data processing to estimate $k_{600}$ and $K_{600}$ from NEON tracer-gas experiments.**

Substantial processing was required to estimate $k_{600}$ and $K_{600}$ from the NEON data (Figure 3). All data used was downloaded from the NEON data portal. Downloads consisted of two NEON data products: Reaeration field and lab collection (DP1.20190.001) and Discharge field collection (DP1.20048.001). Data was from RELEASE-2023, plus nine additional experiments that were provisional but are now included in RELEASE-2024 (NEON, 2023a, b, 2024a, b). On a site-by-site basis, data were merged and formatted using the def.format.reaeration() function from the reaRates R package. This function compiles variables from across the downloaded data into a single data frame. These variables include $Q$, $\bar{w}$, water temperature, station location as distance downstream from the injection point, and $SF_6$ and NaCl or NaBr concentrations for each station during the experiment. The function also applies a salt correction to the $SF_6$ data (e.g., $SF_6$ concentration divided by background-corrected NaCl or NaBr concentrations). The function removes outliers (points more extreme than $1.5 \times IQR$) from the quintuplet $SF_6$ and NaCl or NaBr concentrations for each station, calculates the mean and standard deviation $SF_6$ and NaCl or NaBr concentration for each station, and flags stations as "unmixed" when the coefficient of variation ($CV = \frac{sd}{mean}$) of the replicate $SF_6$ and NaCl or NaBr concentrations is greater than 10%.

**Table 1. Sites and number of experiments**

| Site name | NEON site ID | Velocity experiments (n) | Tracer-gas experiments (n) |
|---|---|---|---|
| Arikaree River | ARIK | 22 | 0 |
| Upper Big Creek | BIGC | 28 | 24 |
| Blacktail Deer Creek | BLDE | 17 | 15 |
| Blue River | BLUE | 21 | 0 |
| Caribou Creek | CARI | 38 | 34 |
| Como Creek | COMO | 37 | 33 |
| Rio Cupeyes | CUPE | 41 | 41 |
| Rio Guilarte | GUIL | 42 | 40 |
| Lower Hop Brook | HOPB | 45 | 40 |
| Kings Creek | KING | 19 | 14 |
| LeConte Creek | LECO | 37 | 31 |



| Lewis Run | LEWI | 46 | 40 |
|---|---|---|---|
| Martha Creek | MART | 33 | 31 |
| Mayfield Creek | MAYF | 43 | 43 |
| McDiffett Creek | MCDI | 32 | 18 |
| McRae Creek | MCRA | 28 | 26 |
| Oksrukuyik Creek | OKSR | 38 | 34 |
| Posey Creek | POSE | 44 | 39 |
| Pringle Creek | PRIN | 14 | 13 |
| Red Butte Creek | REDB | 39 | 39 |
| Sycamore Creek | SYCA | 20 | 19 |
| Teakettle 2 Creek | TECR | 21 | 18 |
| Walker Branch | WALK | 47 | 43 |
| West St Louis Creek | WLOU | 39 | 34 |

Next, SF$_6$ and NaCl or NaBr declines were visualized and quantified for each experiment and initial quality control flags were assigned. The gas.loss.rate.plot() function from the reaRates package was used to visualize and calculate both salt-corrected and uncorrected longitudinal gas loss rates over the length of the study reach ($K_d$). For the sites requiring a salt-correction to account for lateral inflows, mean SF$_6$ mixing ratios at each station were first divided by mean background-corrected NaCl or NaBr concentration for the corresponding station. Station-specific outliers (i.e., values more extreme than first quartile - 1.5*IQR and third quartile + 1.5*IQR) were removed. SF$_6$ concentrations were then log-normalized and $K_d$ was calculated from the resulting linear decline. A quality control flag was assigned to an individual experiment if any of the following criteria applied:

- Visually, the SF$_6$ gas-loss rate over the entire study reach was unduly affected by anomalous SF$_6$ concentrations (potentially indicating contamination, leaked vials, or analytical error)
- SF$_6$, NaCl, or NaBr concentrations increase in a downstream direction between any of the stations (likely indicating incomplete mixing in the water column)
- The salt-corrected $K_d$ was larger than the uncorrected $K_d$ (a salt-correction should correct for overestimation due to lateral inflows, with the reverse potentially indicating contamination or analytical error)

For each experiment, $v$ was calculated from the conductivity time series using the def.calc.trvl.time() function. The function requires that the user manually select points either bracketing the rising limb (for constant rate injections) or the peak concentration (for slug injections). From within the user-selected bracket, the def.calc.trvl.time() function smooths the data using a loess function and then identifies the peak of the breakthrough by either finding where the first derivative is 0 (for a slug injection) or is at its maximum (for a constant rate injection). This function then calculates the breakthrough travel time between the two stations and uses the distance between stations to calculate $v$. Site-specific relationships between $v$ and $Q$ were visualized in log-log space and any anomalous values were reprocessed with the def.calc.trvl.time() to confirm that the user-selected bracketing allowed the function to pick the correct points on the timeseries. Finally, $\bar{z}$ was calculated using the def.calc.trvl.time() function, which divides $Q$ by $v$ and $\bar{w}$.

Two separate approaches were used to estimate $k_{600}$ or $K_{600}$ values from the formatted data. The first approach used the def.calc.reaeration() function to multiply $K_d$ for each individual experiment by the corresponding $v$ and $\bar{z}$ values (equation 2) to estimate $k_{SF_6}$, which were then converted to $k_{600}$ (equation 3). The resulting $k_{600}$ estimates were converted to $K_{600}$ by dividing by water depth. This approach is subsequently referred to as the un-pooled, frequentist approach and is



included in this data descriptor because it represents the current, prevailing approach for processing this type of data, is straightforward to implement, and represents the output of the def.calc.reaeration() function included in the reaRate package.

The second approach used Bayesian multilevel models that pooled experiments from the same site across releases. The models, coded in the Stan probabilistic programming language, used for this approach are also included in the reaRate package. A Bayesian approach provides flexibility in specifying models that consider repeat experiments at a site and current theory surrounding gas exchange. Bayesian inference allowed partial pooling of $k_{600}$ estimates across releases in any one stream. Partial pooling reduces the error in any one estimate of $k_{600}$, and shrinks all $k_{600}$ estimates to the site-level mean (as conditioned on discharge) if error in measuring $SF_6$, NaCl, and/or NaBr is high.

The Bayesian approach included error at two levels. First, the models pooled $k_{600}$ estimates across releases from the same site to estimate $K_d$ from normalized $SF_6$ concentrations (both salt-corrected and uncorrected). For this step, the relationship between the $SF_6$ loss rate and the product of $K_d$ and reach length was assigned a prior normal distribution with a normally distributed sigma (0, 0.2) and intercept (0, 0.1). We fully pooled the intercept with a strong prior near 0 because all $SF_6$ concentrations (i.e., measurements from Stations 1 - 4) were normalized to the $SF_6$ concentration at Station 1; this

approach means that the intercept should be near 1, or 0 when logged. Thus, the model fit can be described as variable-slope, fixed-intercept linear regression. Second, the models pooled the estimates of $k_{600}$ and $K_{600}$ by $Q$, using linear relationships between $Q$ and $k_{600}$ or $K_{600}$. Priors were assigned for both the slope and the intercept based on predictions from an existing scaling model (equation 4 in Raymond et al., 2012). These priors were given large variances when possible (i.e., 10 for the intercept and 1 for the slope) to allow for divergence from the model predictions. However, at sites with a

limited number of experiments (e.g., PRIN and KING), we used smaller variances to allow the model to converge. The site-specific priors used are listed in Table S1.

    The two levels described can be referred to as a within-release model and an among-release model. The within-release model was a log-transformed (base e) exponential model. We log transformed because $SF_6$ is always positive (ambient = 0) and because errors in the measurement of $SF_6$ can scale with the magnitude of the concentration.

$$log(S_{i,j}) = log(S_0) - K_{D,j}x_{i,j} + \varepsilon_{i,j}$$
and
$$\varepsilon_{i,j} \sim normal(0, \sigma_{release}),$$

where $S_{i,j}$ is the $SF_6$ concentration normalized to the concentration at Station 1 for any one release (sample $i$ in release $j$); $S_0$ is the normalized $SF_6$ concentration at Station 1; $x_{i,j}$ is the distance downstream to which the normalized concentration

corresponds; and $\varepsilon_{i,j}$ is a normally distributed random variable with $\mu = 0$ and $sd = \sigma_{release}$. We then converted $K_{D,j}$ to gas exchange velocity $k_{SF_6,j}$ using equation 2 and normalized to $k_{600,j}$ using equation 3. The among-release model included a linear model predicting the parameter $k_{600,j}$ as a linear function of discharge:
$$log(k_{600,j}) = a + b \times log(Q_j) + \varepsilon_j$$
and
$$\varepsilon_j \sim normal(0, \sigma_{stream}),$$

where $a$ and $b$ are the intercept and slope parameters of a model regressing $log(k_{600,j})$ and $log(Q_j)$, where $Q_j$ is the discharge during any one release and $\sigma_{stream}$ is the residual variation. We also fit linear second-level models with $log(K_{600,j})$, where $K_{600,j}$ is the per time gas exchange rate.

    We fit models using Stan in the RStan package in R (Stan Development Team, 2023). The models were run for at

least 5,000 iterations over four chains. Models were assessed according to the number of divergent transitions, the effective sample size (ESS) for each estimated parameter (>1,000), and posterior predictive checks with the shinystan R package (Gabry et al., 2023). In addition, the model fits for each experiment were visually assessed (Figure S1-S2). Finally, the median estimates for $k_{600}$ and $K_{600}$ were visualized in $log(k_{600})$-$log(Q)$ or $log(K_{600})$-$log(Q)$ space, respectively, along with 1,000 MCMC estimates of the $log(k_{600})$-$log(Q)$ or $log(K_{600})$-$log(Q)$ relationship, respectively. If an estimate of $k_{600}$ and

$K_{600}$ fell outside the overall $Q$ relationship and if that experiment's model fits showed signs of being unduly influenced by unrealistic gas loss patterns (e.g., very little decline indicating the study reach was too short, an abrupt decline indicating improper mixing), the experiment was assessed for the QC flags described above. If a QC flag had previously been assigned, then that experiment was removed (e.g., it was identified as an outlier and could be attributed to experimental error), and the model was rerun without that experiment.



## 2.4 Recommended estimates

The processing pipeline outlined above in section 2.3 resulted in both un-pooled frequentist and Bayesian estimates of $k_{600}$ and $K_{600}$, both with and without salt corrections. We include outputs from all four approaches in the dataset for completeness and to allow future users to decide which estimates best fit their needs and to compare the two approaches. The complete dataset is available in the gasExchange_ds.csv file (Aho et al., 2024).

During data processing, we found that the NaCl and NaBr concentration data also could introduce errors and uncertainties into our estimates of $k_{600}$ and $K_{600}$. For instance, background concentrations at a single station were occasionally so high that contamination was the likely explanation. Further, sometimes samples taken during the constant-rate injection could vary across the reach in unpredictable ways (e.g., increases across the reach, random peaks along the reach instead of the expected stable, declining concentrations), which was likely the result of incomplete mixing with the water column. In many cases, the quality of the salt-corrected $SF_6$ data precluded Bayesian-model convergence. Through assessing the gas loss plots and salt concentration declines for all experiments and the failures to produce model convergence for salt-corrected data, we determined that salt corrections had the strong potential to either introduce errors into, or preclude, estimates of $k_{600}$ and $K_{600}$. Therefore, we suggest only using salt-corrected data when clearly necessary (e.g., obvious lateral inflow) and possible in terms of data quality and model convergence. We determined that salt correction was important for five sites: Como Creek (COMO), Rio Cupeyes (CUPE), Rio Yahuecas (GUIL), Martha Creek (MART), and Walker Branch (WALK). Notably, several of these sites have lateral inflows within the study reach under certain hydrologic conditions, which explains the necessity for the salt correction. For completeness, our dataset includes estimates for both uncorrected and salt-corrected $k_{600}$ and $K_{600}$ when a salt-correction is possible.

In addition to the complete dataset of all estimates (i.e., estimates from both frequentist and Bayesian approaches for both uncorrected and salt-corrected data), we also include a curated dataset of recommended estimates of $k_{600}$ and $K_{600}$. These recommended values are called "best_k600_mPerDay" and "best_K600_perDay" in the gasExchange_ds file. In all cases, the curated selection uses the Bayesian estimates because pooling across experiments and the use of informative priors resulted in more meaningful estimates than the non-Bayesian approach. The choice of whether we recommend an uncorrected or salt-corrected estimate stems from examining the relationships between uncorrected or salt-corrected estimates (Figure S3).

# 3 Data description

## 3.1 Hydraulics

Our processing pipeline included calculating hydraulic parameters ($v$ and $\bar{z}$) for each NaCl injections and measurements of $Q$ and $\bar{w}$. These variables ($v$, $\bar{z}$, $Q$, and $\bar{w}$) for each NaCl release are available in the hydraulics_ds.csv file (Aho et al., 2024). Here, we present those data in terms of at-a-station hydraulic geometries, which are commonly used to quantify reach-specific relationships between $Q$ and $\bar{w}$, $\bar{z}$, and $v$ (Leopold and Maddock, 1953):

$$\bar{w} = aQ^b \quad (6)$$
$$\bar{z} = cQ^f \quad (7)$$
$$v = kQ^m \quad (8)$$

In log-log space, these exponential relationships become linear relationships (e.g., $\bar{w} = aQ^b$ becomes $log(\bar{w}) = log(a) + b \times log(Q)$), where the exponent is the slope log-linear relationship. Future users of NEON data can use these relationships along with discharge values from either the Continuous discharge data product (DP4.00130.001) or the Discharge field collection (DP1.20048.001) to approximate $\bar{w}$, $\bar{z}$, and $v$ for NEON streams.

The hydraulic relationships are illustrated (Figures 4-6, Table 2). These geometries leverage field measurements of $\bar{w}$ (n=783) and $Q$ (n=601), estimates of $v$ (n=618) from NaCl injections, and estimates of $\bar{z}$ (n=581) calculated from $Q/v\,\bar{w}$. In general, $\bar{z} - Q$ and $v - Q$ relationships were the strongest, with all but three relationships having $R^2 > 0.5$ and relatively narrow 95% confidence intervals around the coefficients from these relationships (Table 2). The $\bar{w} - Q$ relationships are the weakest; 9 of the 24 sites have $R^2 < 0.5$ and large 95% confidence intervals (Table 2). The $\bar{w} - Q$ relationships may be weaker because our width estimates represent an average of 30 measurements across the ~ 500-m study reach. It is possible that this across-reach averaging contributes to the weaker relationships with $Q$ and perhaps the relationships would be



stronger if the measurement was only taken at the same location as the $Q$ measurement. However, this single point approach would be less compatible with $v$ measured of the entire reach and would alter the resulting calculations of $\bar{z}$.

We assess the quality of our hydraulic parameters by examining the product of the constants ($a \times c \times k$) and sum of the exponents ($b + f + m$) for unity on a site-by-site basis. These unity relationships stem from the fact that $Q = wzv$

300 (Leopold and Maddock, 1953). The products of the constants ranged from 0.93 to 1.04 and averaged 1.00; the site-specific sum of the exponents ranged from 0.96 to 1.01 and averaged 1.00. There was one instance where the product of the constants deviated more than 5% from unity (0.93, PRIN). Pringle Creek (PRIN) is a semi-arid, intermittent stream in Texas, and so the deviation from unit may stem from logistical difficulties in measuring low and non-perennial stream flows (Seybold et al., 2023; Shanafield et al., 2021). However, the remainder of the sites had both products of constants and sums

305 of exponents within 5% of unity.

We assess the representativeness of our hydraulic parameters by comparing to the literature values for the exponents. Previous studies have shown large ranges for all three exponents, with ranges spanning $0 - 0.6$ for $b$, 0-0.8 for $f$, and 0-0.8 for $m$ (Park, 1977; Rhodes, 1977). In addition, exponents have not been shown to vary predictably with region or climate (Park, 1977), but rather with channel geometry (Ferguson, 1986). Our parameters fall within published ranges and

310 our average values for each exponent ($b = 0.11$, $f = 0.33$, $m = 0.56$) are similar to averages in other studies that span many streams ($b = 0.14$, $f = 0.30$ (Morel et al., 2020); $b = 0.12$, $f = 0.37$, $m = 0.51$ (Dingman and Afshari, 2018)). In sum, the hydraulics dataset and associated hydraulic-geometry relationships presented here can be used to characterize $\bar{w}$, $\bar{z}$, and $v$ for NEON streams.



Figure 4. At-a-station hydraulic geometries for relationship between $Q$ and $w$. The exponent $b$ of the power law relationship $\bar{w} = aQ^b$ is the linear slope of the relationship in log-log space and is denoted on each subplot.





Figure 5. At-a-station hydraulic geometries for relationship between $Q$ and $\bar{z}$. The exponent $f$ of the power law relationship $\bar{z} = cQ^f$ is the linear slope of the relationship in log-log space and is denoted on each subplot.



Figure 6. At-a-station hydraulic geometries for relationship between $Q$ and $v$. The exponent of the power law relationship $v = kQ^m$ is the linear slope of the relationship in log-log space and is denoted on each subplot.

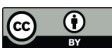



Table 2. Coefficients and exponents from at-a-station hydraulic geometries. The 95% confidence intervals for each coefficient and exponents are shown. In addition, the R² values from the log-linear relationships are also presented.

| Site | $w = aQ^b$ | | | | | $\bar{z} = cQ^f$ | | | | | $v = kQ^m$ | | | | |
|---|---|---|---|---|---|---|---|---|---|---|---|---|---|---|---|
| | a | 95% CI | b | 95% CI | R² | c | 95% CI | f | 95% CI | R² | k | 95% CI | m | 95% CI | R² |
| ARIK | 6.7 | (3.9,11) | 0.04 | (-0.11,0.19) | 0.03 | 0.18 | (0.06,0.56) | 0.15 | (-0.17,0.48) | 0.08 | 0.84 | (0.24,2.9) | 0.80 | (0.46,1.2) | 0.66 |
| BIGC | 4.8 | (4,6.5) | 0.07 | (0.05,0.08) | 0.77 | 0.30 | (0.22,0.4) | 0.27 | (0.18,0.37) | 0.60 | 0.72 | (0.55,0.95) | 0.67 | (0.58,0.76) | 0.91 |
| BLDE | 4.3 | (3.9,4.6) | 0.14 | (0.1,0.17) | 0.78 | 0.30 | (0.28,0.32) | 0.34 | (0.31,0.38) | 0.97 | 0.79 | (0.75,0.83) | 0.52 | (0.49,0.55) | 0.99 |
| BLUE | 23 | (22,24) | 0.01 | (-0.03,0.06) | 0.06 | 0.19 | (0.17,0.22) | 0.53 | (0.4,0.67) | 0.91 | 0.23 | (0.21,0.25) | 0.45 | (0.35,0.56) | 0.92 |
| CARI | 3.3 | (3,3.6) | 0.10 | (0.01,0.18) | 0.16 | 0.50 | (0.45,0.55) | 0.49 | (0.39,0.59) | 0.80 | 0.61 | (0.58,0.64) | 0.41 | (0.37,0.45) | 0.94 |
| COMO | 2.9 | (2.6,3.1) | 0.11 | (0.09,0.13) | 0.82 | 0.51 | (0.41,0.63) | 0.42 | (0.38,0.47) | 0.92 | 0.68 | (0.57,0.82) | 0.47 | (0.43,0.51) | 0.95 |
| CUPE | 6.9 | (6.3,7.5) | 0.13 | (0.1,0.16) | 0.73 | 0.32 | (0.22,0.47) | 0.55 | (0.42,0.68) | 0.72 | 0.45 | (0.32,0.63) | 0.32 | (0.2,0.44) | 0.52 |
| GUIL | 7 | (6.1,7.9) | 0.11 | (0.04,0.18) | 0.26 | 0.45 | (0.39,0.53) | 0.36 | (0.28,0.45) | 0.73 | 0.32 | (0.28,0.36) | 0.53 | (0.46,0.6) | 0.90 |
| HOPB | 6.9 | (6.6,7.3) | 0.13 | (0.11,0.15) | 0.89 | 0.24 | (0.18,0.31) | 0.29 | (0.2,0.39) | 0.64 | 0.61 | (0.47,0.79) | 0.58 | (0.49,0.66) | 0.89 |
| KING | 6.5 | (5.7,7.4) | 0.12 | (0.08,0.17) | 0.76 | 0.28 | (0.12,0.67) | 0.23 | (-0.1,0.55) | 0.20 | 0.54 | (0.24,1.2) | 0.65 | (0.35,0.95) | 0.70 |
| LECO | 8 | (7.4,8.6) | 0.09 | (0.05,0.13) | 0.47 | 0.32 | (0.28,0.37) | 0.40 | (0.34,0.47) | 0.87 | 0.39 | (0.35,0.43) | 0.51 | (0.45,0.56) | 0.93 |
| LEWI | 4.2 | (3.9,4.5) | 0.08 | (0.06,0.11) | 0.51 | 0.33 | (0.26,0.42) | 0.33 | (0.25,0.4) | 0.66 | 0.72 | (0.56,0.93) | 0.59 | (0.51,0.67) | 0.84 |
| MART | 6.4 | (6,6.8) | 0.14 | (0.12,0.16) | 0.89 | 0.24 | (0.22,0.26) | 0.26 | (0.22,0.3) | 0.90 | 0.65 | (0.6,0.7) | 0.58 | (0.55,0.61) | 0.99 |
| MAYF | 5.9 | (5.4,6.3) | 0.13 | (0.1,0.17) | 0.68 | 0.53 | (0.42,0.66) | 0.30 | (0.17,0.42) | 0.50 | 0.32 | (0.27,0.4) | 0.57 | (0.46,0.68) | 0.83 |
| MCDI | 5.6 | (4.7,6.6) | 0.06 | (0,0.12) | 0.22 | 0.38 | (0.14,1) | 0.23 | (-0.11,0.57) | 0.13 | 0.46 | (0.17,1.3) | 0.70 | (0.35,1) | 0.57 |
| MCRA | 6.8 | (6.1,7.5) | 0.09 | (0.05,0.12) | 0.61 | 0.22 | (0.19,0.27) | 0.36 | (0.29,0.42) | 0.90 | 0.66 | (0.58,0.75) | 0.56 | (0.52,0.61) | 0.98 |
| OKSR | 7.3 | (7.2,7.5) | 0.14 | (0.12,0.16) | 0.87 | 0.32 | (0.31,0.33) | 0.22 | (0.18,0.25) | 0.85 | 0.42 | (0.41,0.43) | 0.64 | (0.62,0.67) | 0.99 |
| POSE | 9.7 | (8.1,12) | 0.11 | (0.07,0.15) | 0.46 | 0.19 | (0.15,0.25) | 0.39 | (0.33,0.45) | 0.84 | 0.54 | (0.45,0.66) | 0.50 | (0.46,0.54) | 0.94 |
| PRIN | 6.7 | (5.1,8.9) | 0.15 | (0.04,0.25) | 0.61 | 0.33 | (0.17,0.61) | 0.31 | (0.07,0.56) | 0.57 | 0.42 | (0.24,0.72) | 0.50 | (0.26,0.73) | 0.79 |
| REDB | 3.5 | (3.1,4) | 0.10 | (0.06,0.14) | 0.49 | 0.30 | (0.26,0.35) | 0.33 | (0.28,0.39) | 0.86 | 0.94 | (0.84,1.1) | 0.56 | (0.53,0.6) | 0.97 |
| SYCA | 6.8 | (5.2,8.9) | 0.21 | (0.11,0.31) | 0.65 | 0.22 | (0.18,0.27) | 0.36 | (0.29,0.44) | 0.91 | 0.67 | (0.48,0.93) | 0.43 | (0.3,0.55) | 0.80 |
| TECR | 3.3 | (2.9,3.8) | 0.13 | (0.09,0.16) | 0.79 | 0.27 | (0.19,0.38) | 0.23 | (0.14,0.33) | 0.62 | 1.10 | (0.85,1.5) | 0.64 | (0.57,0.71) | 0.95 |
| WALK | 4.3 | (3.8,4.9) | 0.05 | (0.02,0.08) | 0.43 | 0.16 | (0.09,0.27) | 0.27 | (0.16,0.39) | 0.55 | 1.50 | (0.92,2.4) | 0.67 | (0.57,0.78) | 0.90 |
| WLOU | 3.1 | (2.8,3.3) | 0.11 | (0.09,0.14) | 0.74 | 0.35 | (0.31,0.39) | 0.32 | (0.29,0.36) | 0.91 | 0.93 | (0.85,1) | 0.56 | (0.53,0.58) | 0.98 |





### 3.2 $k_{600}$ and $K_{600}$ estimates

As described above, $k_{600}$ was estimated in two ways: 1) via an unpooled frequentist approach using the def.calc.reaeration() function to estimate $k_{600}$ independently for each experiment and 2) via a partially pooled Bayesian approach that partially pooled experiments from the same site according to $Q$. Both approaches were attempted for raw SF$_6$ concentrations and salt-corrected SF$_6$ concentrations. Salt-corrected SF$_6$ concentrations are only recommended for the five sites mentioned above (COMO, CUPE, GUIL, MART, WALK). All estimates are available in the gasExchange_ds.csv file (Aho et al., 2024).

The relationship between the partially pooled and unpooled estimates (with salt-correction when appropriate) are shown in Figure 7. Any points falling above the 1:1 line are instances when partial pooling resulted in a lower estimate than no pooling, and vice versa. Overall, there are instances where partially pooled estimates are both higher and lower than un-pooled estimates, suggesting that partial pooling successfully regularized estimates. This shrinkage is more obvious when both estimates are plotted against $Q$ (Figure 8). We made recommendations (best_k600_mPerDay and best_K600_perDay in the gasExchange_ds.csv) of which estimates to use; this curated dataset includes only Bayesian estimates, and a salt-correction was only recommended for the five sites where it was possible and necessary.

There are 339 estimates of $k_{600}$ and $K_{600}$ included in our recommended dataset (Figure 9, Table 3). The number of estimates per site range from four (Kings Creek [KING] and Pringle Creek [PRIN]) to 29 (Posey Creek [POSE]). The issueLog.csv file documents the reason why 340 experiments did not successfully produce gas exchange estimates. These reasons are grouped into broad categories and summarized in Table S2. It is possible that some of the experiments that we removed could produce an estimate of $k_{600}$ and $K_{600}$, (e.g., if there was incomplete mixing at the first station, one could remove this station and only estimate $k_{600}$ and $K_{600}$ for stations two to four). However, this type of selective cleaning would have resulted in less comparable estimates (e.g., changing the length of the study reach), so we chose to include only the most comparable and high-quality experiments in this dataset.

The values for recommended estimates of $k_{600}$ ranged from 0.1 m d$^{-1}$ to 142 m d$^{-1}$. LeConte Creek (LECO) had the highest mean $k_{600}$ (mean ± sd: 53 ± 35 m d$^{-1}$) while Pringle Creek (PRIN) had the lowest mean $k_{600}$ (mean ± sd: 1.1 ± 0.1 m d$^{-1}$). Lower Hop Brook (HOPB) had the widest spread of $k_{600}$ values, with estimates ranging almost two orders of magnitude (1.5 - 124 m d$^{-1}$) while Pringle Creek (PRIN) had the smallest spread, with estimates only ranging from 0.9 to 1.2 m d$^{-1}$. The values for recommended estimates of $K_{600}$ range from 1.3 to 481 d$^{-1}$. Like for $k_{600}$, LeConte Creek (LECO) had the highest mean $K_{600}$ (mean ± sd: 276 ± 94 d$^{-1}$) while Pringle Creek (PRIN) had the lowest mean $K_{600}$ (mean ± sd: 7.0 ± 1.9 d$^{-1}$). Also, Lower Hop Brook (HOPB) had the widest spread of $K_{600}$ values, with estimates ranging from 30 to 368 d$^{-1}$ while Pringle Creek (PRIN) had the smallest spread, with estimates ranging from 5.3 to 9.3 d$^{-1}$. These ranges, in part, reflect the various ranges of $Q$ captured at each site (Table S3). HOPB was among the sites with the largest range of $Q$ captured, while PRIN was among the sites with the smallest $Q$ range captured (Table S3). Overall, this large compilation of $k_{600}$ and $K_{600}$ estimates indicate high variability both across and within sites.

Finally, to allow future users to scale $k_{600}$ and $K_{600}$ with $Q$, we include both the slope and intercept for the $k_{600} - Q$ and $K_{600} - Q$ relationships (Table 4) and the stanfit objects for the Bayesian models (Other Entities in the data release). The slope and intercept will allow future users a straightforward way to scale $k_{600}$ or $K_{600}$ as a function of $Q$ at each site. The stanfit objects, on the other hand, will allow future users to sample from the posterior distribution of slope and intercept to assess uncertainty in the scaling relationships.



Figure 7. Comparison of partially pooled Bayesian and unpooled frequentist estimates of $k_{600}$. Black 1:1 lines overlay the points for reference.





Figure 8. Relationship between $Q$ and partially pooled Bayesian and unpooled frequentist estimates of $k_{600}$. The unpooled estimates are shown in green, with a green regression line with 95% confidence intervals, while the partially pooled estimates are shown in purple with a purple regression line with 95% confidence intervals.

380



Figure 9. Box plots of the test estimates of a) $k_{600}$ and b) $K_{600}$ by site. Boxes represent the median and interquartile range (IQR), whiskers mark the lesser of first (third) quartile – (+) 1.5 x IQR or minimum/maximum, and points denote outliers more extreme than first (third) quartile – (+) 1.5 x IQR.

Table 3. Mean, standard deviation (s.d.), minimum, maximum, and count for $k_{600}$ (m d$^{-1}$) and $K_{600}$ (d$^{-1}$) estimates by site.

| Site | n | $k_{600}$ (m d$^{-1}$) | | | | $K_{600}$ (d$^{-1}$) | | | |
|------|---|------|------|-----|-----|------|------|-----|-----|
|      |   | mean | s.d. | min | max | mean | s.d. | min | max |
| BIGC | 22 | 2.1 | 0.9 | 0.7 | 4.1 | 15.6 | 7.8 | 6.2 | 39.9 |
| BLDE | 11 | 5.9 | 6.2 | 1.7 | 23.4 | 33 | 19.9 | 12.1 | 83.1 |
| CARI | 16 | 9.5 | 8.3 | 3.4 | 35.6 | 31.7 | 32.5 | 6.1 | 140.9 |
| COMO | 17 | 26.8 | 32.1 | 2.7 | 94.8 | 182.7 | 91.9 | 66.6 | 377.3 |
| CUPE | 26 | 6.5 | 3.1 | 3.1 | 15.8 | 104.7 | 33.5 | 47.3 | 190.1 |

| GUIL | 15 | 9.3 | 6 | 3.6 | 22.4 | 39.7 | 21.7 | 15 | 83.1 |
|------|----|-----|----|-----|-------|-------|------|------|-------|
| HOPB | 19 | 15 | 27.8 | 1.6 | 123.6 | 90.7 | 80.4 | 29.5 | 367.6 |
| KING | 4 | 4.3 | 2 | 2.8 | 7.2 | 34.5 | 25.6 | 12.2 | 62.4 |
| LECO | 10 | 53.2 | 35.4 | 21.4 | 142.2 | 275.7 | 94.4 | 156 | 481.4 |
| LEWI | 21 | 2.2 | 1 | 0.9 | 4.6 | 19.2 | 7.1 | 9.1 | 34.5 |
| MART | 7 | 1.2 | 1.2 | 0.1 | 3.4 | 12.3 | 13.3 | 1.3 | 35.7 |
| MAYF | 7 | 12.3 | 11 | 2.8 | 34.3 | 39 | 40.5 | 9.3 | 126.9 |
| MCDI | 6 | 4.8 | 0.8 | 3.5 | 5.7 | 26.8 | 16.7 | 10.2 | 53.2 |
| MCRA | 16 | 10.6 | 9.9 | 1.2 | 33.9 | 87.8 | 46.5 | 27.6 | 185.9 |
| OKSR | 21 | 4.8 | 5.8 | 1.4 | 29.4 | 15.4 | 14.4 | 4.2 | 74 |
| POSE | 29 | 3.8 | 2.6 | 0.7 | 11 | 108.6 | 35.5 | 48.9 | 187.4 |
| PRIN | 4 | 1.1 | 0.1 | 0.9 | 1.2 | 7.2 | 1.7 | 5.7 | 9.2 |
| REDB | 25 | 14.1 | 11.3 | 3 | 53.9 | 102.9 | 40.2 | 42.4 | 181.4 |
| SYCA | 11 | 2.3 | 2.3 | 0.3 | 8 | 30 | 23.8 | 3.7 | 64.6 |
| TECR | 15 | 12 | 8.4 | 3.2 | 28.7 | 93.5 | 42 | 33.9 | 181.9 |
| WALK | 9 | 2.1 | 1.1 | 0.1 | 3.6 | 59.9 | 22.2 | 6.6 | 82.2 |
| WLOU | 28 | 31.6 | 26 | 9.6 | 112.9 | 199.6 | 83.4 | 93.8 | 410 |

395    Table 4. Coefficients and exponents for k600-Q and K600-Q relationships. Estimates and standard deviations (SD) are given.

| | $k_{600}= aQ^b$ | | | | $K_{600}=aQ^b$ | | | |
|------|----------|------|----------|------|----------|------|----------|------|
| | a | | b | | a | | b | |
| Site | estimate | SD | estimate | SD | estimate | SD | estimate | SD |
| BIGC | -0.31 | 1.55 | 0.11 | 0.18 | 3.55 | 1.57 | -0.11 | 0.19 |
| BLDE | -2.11 | 2.27 | 0.38 | 0.24 | 2.88 | 2.15 | 0.05 | 0.23 |
| CARI | 0.06 | 3.51 | 0.19 | 0.34 | 6.50 | 3.69 | -0.32 | 0.36 |
| COMO | -2.30 | 0.34 | 0.69 | 0.05 | 3.17 | 0.32 | 0.27 | 0.04 |
| CUPE | -2.74 | 0.54 | 0.54 | 0.07 | 4.84 | 0.92 | -0.03 | 0.11 |
| GUIL | -0.51 | 2.84 | 0.27 | 0.30 | 4.50 | 2.58 | -0.10 | 0.28 |
| HOPB | -3.83 | 0.64 | 0.67 | 0.07 | 0.94 | 0.48 | 0.39 | 0.05 |
| KING | -1.97 | 2.47 | 0.38 | 0.25 | 0.06 | 2.76 | 0.36 | 0.30 |
| LECO | -5.68 | 2.21 | 0.97 | 0.23 | 0.66 | 1.85 | 0.50 | 0.19 |
| LEWI | -2.18 | 1.78 | 0.36 | 0.23 | 3.48 | 1.78 | -0.07 | 0.23 |
| MART | -0.55 | 4.34 | 0.03 | 0.55 | 3.82 | 4.34 | -0.23 | 0.54 |
| MAYF | -3.44 | 5.44 | 0.59 | 0.57 | -0.04 | 5.46 | 0.35 | 0.57 |





| MCDI | 0.40 | 2.02 | 0.13 | 0.24 | 0.29 | 4.06 | 0.33 | 0.48 |
|------|------|------|------|------|------|------|------|------|
| MCRA | -3.07 | 1.06 | 0.57 | 0.12 | 2.37 | 0.93 | 0.22 | 0.11 |
| OKSR | -1.41 | 1.94 | 0.25 | 0.18 | 2.30 | 2.04 | 0.02 | 0.19 |
| POSE | -2.90 | 0.36 | 0.60 | 0.05 | 3.21 | 0.37 | 0.21 | 0.05 |
| PRIN | -2.69 | 0.88 | 0.30 | 0.10 | 2.37 | 0.97 | -0.05 | 0.11 |
| REDB | -2.99 | 0.70 | 0.63 | 0.08 | 1.97 | 0.60 | 0.30 | 0.07 |
| SYCA | -0.73 | 2.67 | 0.14 | 0.32 | 5.15 | 2.77 | -0.25 | 0.33 |
| TECR | -1.47 | 0.63 | 0.48 | 0.08 | 2.61 | 0.71 | 0.23 | 0.09 |
| WALK | -4.03 | 3.48 | 0.73 | 0.56 | 0.96 | 3.09 | 0.48 | 0.50 |
| WLOU | -2.15 | 0.30 | 0.64 | 0.04 | 2.48 | 0.31 | 0.33 | 0.04 |

## 4 Code and data availability

The Reaeration and Discharge data are available for download from the NEON data portal (http://data.neonscience.org). The reaRates R package is available at https://doi.org/10.5281/zenodo.12786089 (Cawley et al., 2024). The dataset of hydraulic parameters and gas exchange estimates is available from the Environmental Data Initiative: https://doi.org/10.6073/pasta/8faa6ed1b1d8d1e7ad6c9e897bcacc49 (Aho et al., 2024).

## 5 Conclusions

Here, we present 339 estimates of gas exchange velocity ($k_{600}$) and gas exchange rate ($K_{600}$) from 22 NEON streams. To our knowledge, this dataset is the largest compilation of gas-exchange estimates from standardized tracer-gas experiments (i.e., standardized methods across all experiments and sites) published to date. Given the wide geographic spread of NEON streams (e.g., spanning from Puerto Rico to Alaska), this dataset will enable understanding gas exchange across biomes. In addition, in our estimation process, we leverage new Bayesian multilevel models that allow estimates of gas exchange to be partially pooled for each sites according to $Q$; this modeling approach can be applied to future studies where repeat tracer-gas experiments are conducted to characterize gas exchange as a function of $Q$. Here, we also present relationships between discharge and gas exchange (i.e., $k_{600} - Q$ and $K_{600} - Q$) from these models that will allow future users to scale $k_{600}$ or $K_{600}$ as a function of $Q$ in NEON streams.

Regarding hydraulics, we present hydraulic geometries for 24 NEON streams. These geometries leverage field measurements of $\bar{w}$ and $Q$, and estimate $v$ and $\bar{z}$. The coefficients and exponents from the at-a-station hydraulic geometries are presented, and can be used in the future, along with Continuous discharge (DP4.00130.001) to estimate $\bar{w}$, $v$, and $\bar{z}$ at NEON streams. In sum, this large dataset could allow for quantification of both within- and across-reach variability in hydraulics and gas exchange in streams, which could be useful to modeling stream metabolism, greenhouse gas emissions, and other biogeochemical fluxes in NEON streams. In addition, this dataset may facilitate the development of new predictive models of gas exchange in small streams.

## 6 Author contribution

Conceptualization: KA, KC, RH, ROH, WD, KG; Methodology: KA, KC, RH, ROH with support from all authors; Software: KA, KC, ROH; Writing – Original Draft: KA with support from all authors; Writing – Review and Editing: KA, KC, RH, ROH, WD, KG.





## 7 Competing interests

The authors declare that they have no conflict of interest.

## 8 Acknowledgements

The idea for this dataset initially came from meetings of the National Ecological Observatory Network (NEON) Re-aeration Technical Working Group.

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
