# Peer review of "Gas exchange velocities $(k_{600})$ , gas exchange rates $(K_{600})$ , and hydraulic geometries for streams and rivers derived from the *NEON Reaeration field and lab collection data product* (DP1.20190.001)"

_Earth System Science Data, 2024_

## Author Comment (AC1)

Aho et al. presents a large dataset of gas exchange velocities (k600) and gas exchange rates (K600) along with hydraulic geometries from 22 stream and river sites across the United States. This dataset is based on rigorous field experiments and measurements and is the largest gas exchange dataset I have ever seen. The value of the dataset is to facilitate key research questions such as riverine greenhouse gas (GHG) emissions and river metabolism and help understand the river ecological processes and biogeochemical fluxes. The manuscript is well-organized and straightforward to follow. The field experiments, data collection, data processing, and data description are all described in great details, although more information is needed for some aspects. Generally, I endorse the publication of this work in ESSD. I have the following specific comments need to be addressed.

Thank you for the positive assessments of the dataset and descriptor.

L64: The term "$k600$" and "$K600$" should be defined and/or explained as they appear first time in the main text. Those two parameters look very similar (differ in non-capital and capital K) and they may confuse the readers. An explanation here would be very helpful.

Great suggestion. We added definitions (underlined) to this part of the introduction:
L67-68: The dataset presented here contains estimates of $k_{600}$, or $k$ normalized to a Schmidt number of 600, and $K_{600}$, which is a depth-corrected rate ($K_{600} = k_{600}/\bar{z}$) used in stream metabolism modeling, for 22 streams and $v$, $\bar{z}$, and at-a-station hydraulic geometry for 24 streams (Figure 1).

L76: Why use $SF_6$ as tracer gas? $SF_6$ is a very potent GHG that has 23500 times greater global warming potential than $CO_2$. Such large-scale experiments may cause environmental burdens.

This is an excellent point and one of the reasons that NEON is ceasing tracer-gas experiments at sites with sufficient hydrologic coverage. In the literature, SF6 has been used in tracer-gas experiments because it is not naturally occurring, is nonreactive in the environment, and can be detected at very low concentrations (Cole and Caraco, 1998; Ho et al., 2011; Maurice et al., 2017; Wanninkhof et al., 1985). However, even though these experiments use very small amounts of SF6, it does have a high global warming potential. We added a discussion of this tradeoff to the methods:
L87-90: NEON uses $SF_6$ because it does not occur naturally, is not biologically or chemically reactive, and can be detected at low concentrations (Cole and Caraco, 1998; Ho et al., 2011; Maurice et al., 2017; Wanninkhof et al., 1985); although used in very small amounts in these experiments, $SF_6$ is a potent greenhouse gas and tracer-gas experiments are ceasing at sites with sufficient hydrologic coverage.

L83: How was the tracer gas sample collected? Using headspace equilibrium method? More details are needed.
L87: How was the tracer $SF_6$ samples analyzed?

In response to both comments, we added the following information:
L97-99: Samples for $SF_6$ are collected via headspace equilibration in 60-ml syringes, stored in gas-tight evacuated vials (12-ml, Exetainer), and run on a gas chromatograph with an electron

capture detector (ECD). Samples for NaCl and NaBr are filtered to 0.7 μm, collected in 60-ml HDPE bottles, and run on an ion chromatograph.

L90: Can you plot the discharge hydrograph and marked when the tracer experiments were conducted? Histograms of discharge with tracer experiment marks on them are also acceptable. These graphs, which can be put in the supplement, will clearly show the representativeness of the experiments with flow regimes.

Yes, we added hydrographs with points indicating when k600/K600 and hydraulic geometry was estimated (Figure S4). We also added flow duration to curves to clearly show the representation of flows captured (Figure S5). These new figures are referenced in the text:

L311-312: "The timing of these measurements are denoted on site-specific hydrographs for each site (Figure S4) and their coverage of site-specific $Q$ ranges are illustrated on flow duration curves (Figure S5)."

L376-377: "There are 339 estimates of $k_{600}$ and $K_{600}$ included in our recommended dataset (Figure 9, Table 3) that span large $Q$ ranges at each site (Figure S4-S5)."

L368: Scaling relationships between k600 and hydraulics parameters such as velocity and channel slope would also be very useful for future users to estimate k600. I suggest the authors provided the equations between k600 and velocity and channel slope if that's feasible.

L418: Can you go a step forward and provide these predictive models of gas exchange in this paper? With such comprehensive dataset in-hand, those models should be easy to fit (see the comment above).

In response to both comments, we have recently submitted a separate manuscript to another journal that examines controls on k600, including velocity and slope, using this large dataset. We prepared a separate analysis-focused manuscript because, according to ESSD guidelines, "extensive interpretations of data – i.e. detailed analysis as an author might report in a research article – remain outside the scope of this data journal." Please keep a look out for the companion analysis!

RC2

'Comment on essd-2024-330', Liwei Zhang, 07 Sep 2024

This dataset presents a large compilations of gas exchange related estimates based on gas loss experiments in streams across a range of discharges. It will be an important resource for the aquatic biogeochemistry community in general. Such compilation is necessary in order to better quantify the role and contribution of fluvial systems to greenhouse gas emissions, as well as to better identify current geographic gaps. The paper is structured in a good way that allows readers to follow easily. I have only few suggestions and minor comments in the hope that these can be helpful to the authors. Thanks to the authors for this important contribution to the field, and I'd like to see this work could be published in ESSD.

Thank you for these comments!

Line by Line Comments:

L31: I noticed the authors cited Rocher-Ros et al., 2023, so I wonder if the authors are able to provide some discussion about missing information of ebullitive k for $CH_4$.

Good point. We added clarification around the gas transfer processes captured by these estimates:
L45-46: Here, $k$ refers to exchange of dissolved gases (i.e., diffusive and bubble-mediated gas transfer) and does not capture ebullitive fluxes.

L77: It seems like these experimented streams are in small size. This raises the question of whether the methodologies and conclusions drawn from this study can be reliably extrapolated to the context of big rivers.
Yes, the NEON sites are located on streams and small rivers. We agree that this point is important to highlight for future use of this dataset and have added a few sentences about the size range to the methods:
L82-85: In general, the NEON streams are relatively small; the median watershed size is 11.5 km$^2$ while the mean ± standard deviation watershed size is 27.5 ± 58.5 km$^2$. Walker Branch (WALK) drains the smallest watershed (1.1-km$^2$) while Sycamore Creek (SYCA), an intermittent, desert stream, drains the largest (280-km$^2$). Therefore, this dataset is only representative of streams and small rivers, and not large rivers.

Please indicate the dates or seasons of experiments somewhere in the methods, because these are key information regarding discharge, velocity, water depth, etc.
Good point! We updated the EDI data release (now Version 2 is available on EDI and cited in this manuscript) to include timestamps for each experiment (please see new column "collectDate_UTC" in "hydraulics_v2.csv" and "gasExchange_ds_v2.csv"). We also added hydrographs (Figure S4) that show when experiments were conducted over the multiyear collection period. Finally, thanks to this comment, we created flow duration curves (Figure S5) to illustrate the discharges captured by this dataset.